# A Review on Recent Progress of Glycan-Based Surfactant Micelles as Nanoreactor Systems for Chemical Synthesis Applications

Bahareh Vafakish  and Lee D. Wilson *

Department of Chemistry, University of Saskatchewan, 110 Science Place, Saskatoon, SK S7N 5C9, Canada; bav128@usask.ca
* Correspondence: lee.wilson@usask.ca; Tel.: +1-306-966-2961; Fax: +1-306-966-4730

**Abstract:** The nanoreactor concept and its application as a modality to carry out chemical reactions in confined and compartmentalized structures continues to receive increasing attention. Micelle-based nanoreactors derived from various classes of surfactant demonstrate outstanding potential for chemical synthesis. Polysaccharide (glycan-based) surfactants are an emerging class of biodegradable, non-toxic, and sustainable alternatives over conventional surfactant systems. The unique structure of glycan-based surfactants and their micellar structures provide a nanoenvironment that differs from that of the bulk solution, and supported by chemical reactions with uniquely different reaction rates and mechanisms. In this review, the aggregation of glycan-based surfactants to afford micelles and their utility for the synthesis of selected classes of reactions by the nanoreactor technique is discussed. Glycan-based surfactants are ecofriendly and promising surfactants over conventional synthetic analogues. This contribution aims to highlight recent developments in the field of glycan-based surfactants that are relevant to nanoreactors, along with future opportunities for research. In turn, coverage of research for glycan-based surfactants in nanoreactor assemblies with tailored volume and functionality is anticipated to motivate advanced research for the synthesis of diverse chemical species.

**Keywords:** nanoreactors; glycan-based surfactants; micelle structure; compartmentalization; chemical synthesis

## 1. Introduction

Nanoreactors are tiny containers that can serve as host systems to encapsulate starting materials in an internal cavity volume with nanoscale dimensions [1]. In contrast with bulk solution, nanoreactors contribute to an increased reactant (guest) concentration in the nanoreactor host cavity, where specific orientation factors can affect the reaction progress and product properties [2]. In comparison to reactions in bulk solution media, colloidal nanoreactors represent an area of continued research interest. Compartmentalization of chemical species upon phase transfer from the bulk solution to micellar media accounts for dramatic changes in the reaction rate, besides the physicochemical properties of products such as morphology and molecular weight [3].

The use of water as a solvent offers an alternative reaction media versus harmful and toxic organic solvent systems. However, the low water solubility of organic precursors in aqueous media pose potential limitations for a wider field of application, especially for hydrophobic reactants. In order to advance the use water-based media, site-isolated environments such as amphiphilic organic hosts [4] or nanoreactors represent a potential solution for the dissolution of water-insoluble organic compounds.

Strategies directed at performing chemical reactions in water are becoming increasingly important based on a consideration of the principles of green chemistry. The use of water-based reaction media has inspired researchers to develop environmentally friendly

and sustainable production strategies [5] for the chemical industry to meet the ever-increasing demands of environmental regulatory requirements [6]. In the case of water-insoluble reactants, the use of water as a primary solvent has limitations, where the use of colloidal additives or supercritical conditions offer potential solutions [7,8]. Among these methods, the use of preorganized macromolecular host systems with amphiphilic character such as cyclodextrins [9,10] offer a unique organic microenvironment that favour reactions in aqueous media. In contrast to preorganized host systems, colloidal self-assembly offers a modular approach to address the limited water solubility of reagents. The structure of colloidal systems can be tailored by judicious choice of glycan-based surfactants with variable molecular weight that undergo self-assembly to yield nanoreactor constructs by conventional or reverse micellar media [11].

Many examples of chemical reactions in micro heterogeneous systems are known in nature. A classic example is the biosynthesis of pure *cis*-1,4-polyisoprene isomer in the latex form (dispersed polymer in water) by the rubber tree [12]. Nearly eighty years ago, various attempts to replicate this process led to an industrial scale latex production that employed an emulsion polymerization technique [13]. This was one of the first examples describing the synthesis of chemical products in micellar reaction media [14]. The structure and preparation of these micellar nanoreactors by self-assembly is an ostensibly more facile strategy, as compared with the use of advanced synthetic preorganized host-systems such as metal-organic frameworks (MOFs) [15,16], calixarenes [17,18], pillarenes [19,20], modified cyclodextrins [21,22], cucurbit[n]urils [23,24], zeolites [25,26], and sol–gel porous materials [27,28]. As well, the utility of self-assembled nanoreactors can greatly affect the reaction rate, mechanistic pathway, stereochemistry, molecular weight, distribution, and morphology of the reaction products.

Currently, efforts are directed at finding alternatives such as biobased surfactants from renewable materials to replace conventional synthetic surfactants due to the known environmental toxicity and non-degradability of the latter [29]. The current worldwide demand for surfactants is ca. 16.5 million mega tonnes with an AAGR (average annual growth rate) of 2.6% per year [30]. Obviously, even partial replacement of toxic synthetic surfactants with glycan-based alternatives will have a tremendous impact on the sustainability of the surfactant industry. Glycan-based or carbohydrate-based surfactants are categorized as biodegradable and non-toxic amphiphiles [29,30]. Such types of glycan-based systems can form micelles in aqueous media and yield nanoreactor assemblies similar to conventional synthetic amphiphiles [31]. Though some reports have described the synthesis and physico-chemical properties of glycan-based surfactants [32–34], there are limited literature reviews for these systems that describe their synthesis and micellar properties as nanoreactors.

Hence, this contribution aims to provide a general overview of glycan-based surfactants and their micellization properties over the past decade, along with their utility as micellar nanoreactors for chemical synthesis of diverse products: inorganic compounds, organic materials, polymers, and other selected examples chemical products. To the best of our knowledge, there is no literature review that overlaps with the coverage outlined herein for this area of polysaccharide-based amphiphiles.

### 1.1. Types of Glycan-Based Surfactants

Glycan-based surfactants typically contain a monomer or polysaccharide unit in the amphiphile structure, where the hydrophilic head group is attached to a lipophilic fragment, which imparts surface activity in aqueous media that is similar to conventional synthetic surfactant systems. By changing the hydrophilic head group and length of the tail, the physicochemical properties of these surfactants can be tailored for specific applications. The occurrence of structural variants for glycan-based surfactants with cationic or anionic groups as part of the typical backbone motif or the presence of additional hydrophile head groups and/or additional lipophile fragments in the molecular structure [32]. Numerous types of glycan-based surfactants were recently reported [34–36], wherein a small subset has attracted wide attention. An important feature for industrial application of glycan-

based surfactants is their biodegradability, where the nature of the functional groups in the surfactant structure will affect the biodegradation rate, as evidenced by the greater stability of glucamides to hydrolysis over sorbitan fatty acids [37]. As well, the presence of cationic or anionic groups will decrease the degradation rate [38]. The following discussion will outline the availability of raw materials, facile modality of glycan amphiphile synthesis and the quality of the final products in the sections below.

1.1.1. Span and Tween

Span and Tween are the original registered trade names of sorbitan fatty esters and their ethoxylated forms, respectively, where they represent the most commonly used glycan-based surfactants. Span is prepared by linking a long chain fatty acid to the sorbitan head group, which results upon dehydration of sorbitol [39]. Tween is prepared by ethoxylation of Span in alkaline media, which serve to alter the HLB (hydrophile–lipophile balance), in comparison with the parent compounds [40]. A typical example of the chemical structures of Span and Tween are shown in Scheme 1A. The type of fatty acid attached to the sorbitan head group confers structural features that affect the surface-active properties accordingly, where the typical number of ethylene oxide (EO) units is generally about twenty [33].

**Scheme 1.** Chemical structures of various glycan-based amphiphiles: (**A**) Span and its ethoxylated form, Tween; (**B**) alkyl polyglycoside (APG); (**C**) fatty acid glucamide; and (**D**) a sucrose ester.

1.1.2. Alkyl Polyglycosides

Alkyl polyglycosides (APGs) are typically prepared using a solvent-free direct or indirect acetalization of the hemiacetal hydroxyl group of the glucose with a fatty alcohol. The products that result are a complex mixture containing different types of glucose monomers or oligomers, isomers, and anomers [39]. The chemical structure of APGs is represented in Scheme 1B, where the chain length of the fatty alcohol governs the surface-activity. The characteristic features of APGs include insensitivity of their properties such

as water solubility, compatibility with other surfactants, cloud point, Kraft point, high tolerance to electrolytes, and efficient emulsification.

### 1.1.3. Fatty Acid Glucamides

Fatty acid glucamides are prepared from reductive amination of glucose followed by acylation with a fatty acid [39]. Considering their amide linkage, they are less sensitive to alkaline hydrolysis unlike sorbitan esters. Scheme 1C illustrates the chemical structure of a typical fatty acid glucamide.

### 1.1.4. Sucrose Esters

Sucrose esters are prepared by solvent free esterification or transesterification of molten sucrose with fatty acids [39]. A model chemical structure of a sucrose ester is shown in Scheme 1D. Since sucrose is thermally sensitive and contains different functional groups, the synthesis of such types of glycan-based surfactants is more challenging than others. Typically, a complex product mixture containing monoester, diester, and polyester is produced, where an average degree of substitution (DS) is targeted at DS = 2.5 [34]. The hydrophilic character of sucrose limits the type of fatty acid chain in the structure to be longer than ten carbons ($n = 10$). For the case where a shorter fatty acid is used ($n < 10$), the surface activity of the sucrose ester is lost.

### *1.2. Micellization Properties*

To expand the application of glycan-based surfactants, it is necessary to gain further insight about their micellar self-aggregation and surface-active properties at similar conditions as that for petroleum based surfactants. The study of the aggregation behaviour of glycan-based surfactants, along with their thermodynamic properties will contribute to a greater understanding of their utility as amphiphilic components in complex systems [41].

### 1.2.1. Surface Tension Measurements

Surface tension measurements provide insight related to the cohesive forces at the air–water interface. The equilibrium surface tension can be characterized by use of the Du Nouy ring or Wilhelmy plate methods [42]. From the break in the curve of surface tension ($\gamma$) versus surfactant concentration, the critical micelle concentration (CMC) can be estimated [35]. The initial dependence of the $\gamma$-values decrease as the surfactant concentration increases as amphiphiles are adsorbed at the air–water interface until saturation occurs, where assembly of the surfactant yields micelles thereafter [43,44]. At the CMC, the surface tension remains constant due to the more energetically favourable path of micelle formation instead of transport of surfactant species to the interface (Figure 1) [41,42] Micelle formation is favourable due to removal of hydrophobic fragments and reforming of hydrogen bonds in the water [45,46]. Since there are various types of head groups and tails that can be used for the synthesis of glycan-based surfactants, it is not possible to give a discrete value for the surface tension for glycan-based surfactants. Their equilibrium surface tension and CMC value are generally lower than the corresponding conventional anionic and cationic surfactants [47].

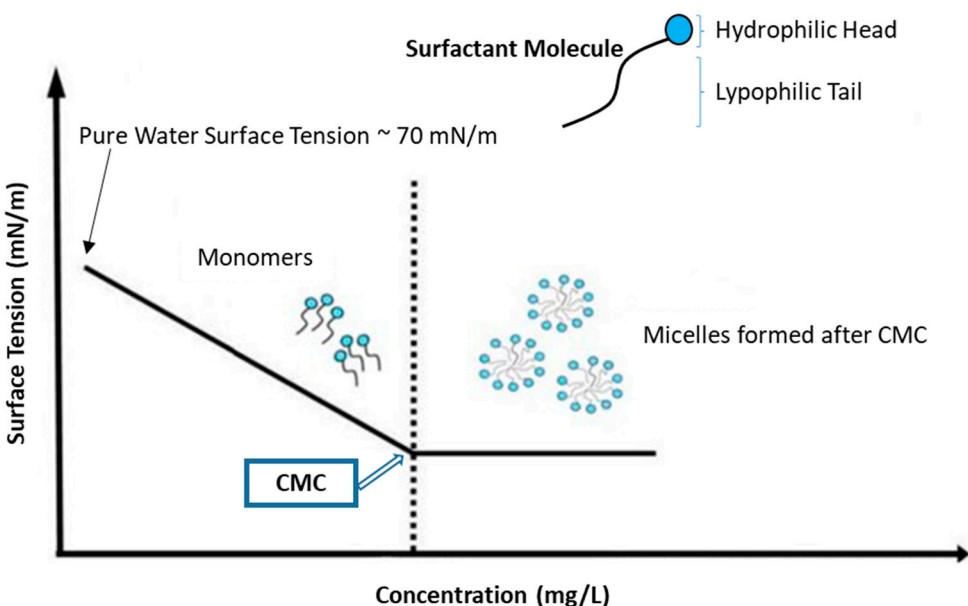

**Figure 1.** Surface tension decreases with increasing the surfactant concentration up to the critical micelle concentration (CMC), where the concentration dependence is nearly constant above the CMC. Redrawn with modification from Ref [44].

### 1.2.2. Particle Size Distribution of Micelles

Dynamic light scattering (DLS) is a technique that may be used to determine the particle size and the size distribution of micelles, which is usually shown as a mono- or bimodal distribution for glycan-based surfactants [48]. The diameter ($d$) or radius ($d/2$) may be determined by DLS or other suitable methods. The hydrodynamic size of the micelle represent an apparently larger quantity versus the actual micelle diameter due to hydration and aggregation effects. The size distribution and mean micelle diameter of Tween 20 determined by DLS is shown in Figure 2 [49].

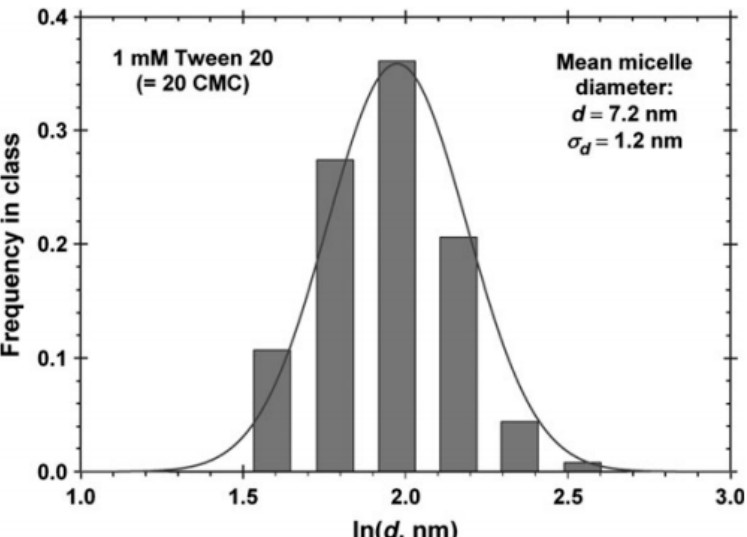

**Figure 2.** Monomodal size distribution of micelles of Tween 20 determined by dynamic light scattering (DLS). The mean micelle diameter ($d$) is 7.2 nm with the standard deviation of 1.2 nm. Reprinted without modification from Ref. [49], with permission from the PCCP owner societies, Published by the RSC (Royal Society of Chemistry).

When laser light encounters micelles, it is scattered, and the intensity of such scattering is proportional to translational diffusion coefficient (TDC). The size of the micelles and their

size distribution are calculated from TDC through the Stokes–Einstein equation [45,46]. The DLS detector is usually located at 90° or 173° relative to the direction of the incident laser light. For some surfactant solutions with low CMC, the scattered light that strikes the detector is relatively low so an attenuator affords more laser light to pass through the sample to maximize the scattered light intensity [50–53].

Micellar size, shape, and the aggregation number often depend on the chain length of the lipophilic tail, hydrophobic head group of the surfactant molecules, and the role of inter- and intramolecular interactions. Self-assembly is also affected by external factors like concentration and nature of the salt in the emulsion, cosolvents, and temperature. For example, by increasing the salt concentration, the hydrodynamic radius increased while temperature has adverse effects because the standard enthalpy for association of such surfactants is exothermic. Table 1 highlights the hydrodynamic radius ($R_h$) of some selected glycan based surfactants systems in water.

**Table 1.** Hydrodynamic diameter ($R_h$) of some glycan-based surfactants in aqueous media.

| Glycan-Based Surfactant | Hydrodynamic Diameter ($R_h$; nm) | Reference |
|---|---|---|
| *N*-C14-lactosamine | 9 [a] | [53] |
| sodium methyl 2-dodecanoyl amido-2-deoxy-6-*O*-sulfo-D-glucopyranoside | 2.2 | [54] |
| sodium methyl 2-hexadecanoyl amido-2-deoxy-6-*O*-sulfo-D-glucopyranoside | 15.8 | [54] |
| *N*-nonanoyl-*N*-methyl-D-glucamine | 2.4 [b] | [55] |
| Stearic Acid Sucrose Monoester | 4.8 | [56] |
| Capric Acid Sucrose Monoester | 3.3 | [56] |
| Span-20, Tween-20 (60–40) with 5% water | 30 [c] | [57] |

[a] The $R_h$ increased to 15 nm by incorporation of starting material into a micellar nanoreactor. [b] The $R_h$ in NaCl (1 M) solution is 2.9 nm and in $T$ = 323 K is 2.2 nm. [c] By increasing the water content (%), the $R_h$ increased to 150 nm.

### 1.2.3. Adsorption Behaviour

The surface excess concentration ($\Gamma_{max}$) describes the excess of adsorption at the air–water interface relative to bulk solution can be estimated according to Equation (1):

$$\Gamma_{max} = -\frac{1}{RT}\left(\frac{d\gamma}{d\ \ln C}\right) \tag{1}$$

$\Gamma_{max}$ is the surface excess concentration (mol/m$^2$), R is the gas constant, $T$ is the absolute temperature, $\gamma$ is the surface tension and C represents the surfactant concentration in water [58]. From the surface excess concentration, the minimum area that each surfactant molecule occupies ($A_{min}$) can be calculated according to Equation (2):

$$A_{min} = \frac{10^{20}}{N\ \Gamma_{max}} \tag{2}$$

$N$ is Avogadro's number and $\Gamma_{max}$ represents the surface excess concentration. The size of head group and tail both affect the magnitude of the minimum area. Since the glycan head groups may form hydrogen bonds with their neighbours, the value of $A_{min}$ for glycan-based surfactants is large, especially for surfactants with long hydrocarbon tails that enhance such hydrophobic effects [59].

The standard change in Gibbs energy of micellization ($\Delta G°_{mic}$) is another important parameter that provides an understanding of micelle formation in aqueous media, which can be calculated by applying the value of $X_{CMC}$ in Equation (3).

$$\Delta G°_{mic} = RT\ln X_{CMC} \tag{3}$$

$R$ and $T$ are defined as in Equation (1), whereas $X_{CMC}$ is the mole fraction of surfactant at the *CMC* [60]. Glycan-based amphiphiles can be treated similar to conventional surfactants, as supported by negative values of $\Delta G°_{mic}$, which indicates a spontaneous

micellization process [61]. It is noteworthy that glycan-based surfactants reveal a more negative $\Delta G°_{mic}$ as compared to conventional surfactants [54]. Effective hydrogen bonds between the sugar head groups, along with hydrophobic effects, contribute to favourable thermodynamics for the micellization process.

Some interfacial properties of various industrially important glycan-based surfactants are listed in Table 2 that include the *CMC*, $\Gamma_{max}$, $A_{min}$, and equilibrium surface tension.

**Table 2.** Interfacial properties of the selected glycan-based surfactants.

| Glycan-Based Surfactant | CMC (mM) | $\Gamma_{max}$ (μmol/mL) | $A_{min}$ (nm²) | $\gamma$ (mN/m) | Reference |
|---|---|---|---|---|---|
| Sorbitan monooleate | 2.3 | 4 | 0.41 | 30 | [61–63] |
| Ethoxylated sorbitan monooleate | 0.20 | 1.4 | 0.12 | 37.5 | [62–64] |
| $C_{10}$ Alkyl polyglycoside | 2.2 | 1.59 | - | 29 | [65–67] |
| $C_{12}$–$C_{14}$ Alkyl polyglycoside | 1.63 | 1.74 | 0.29 | 27 | [65–67] |
| Sucrose mono-decanoate | 0.60 | 17.6 | 0.09 | 26.9 | [68–70] |
| Sucrose monolaurate | 0.45 | 9.1 | 0.18 | 22 | [56,59–61] |
| Dodecanoyl methyl glucamide | 0.41 | - | - | 35.5 | [71,72] |

## 2. Application of Glycan-Based Surfactant's Nanoreactors for Chemical Synthesis

### 2.1. Synthesis of Inorganic Compounds

There are reports on the synthesis of inorganic compounds like platinum, palladium, and rhodium [73], nanoiron [74], nickel oxide [75], and ceramic particles [57] that employ glycan-based surfactant nanoreactors. It has been less than forty years since the pioneering work of Boutonnet et al. in 1982, where they reported the preparation of metallic nanoparticles in an emulsion system [76]. The field of metal nanoparticles has witnessed enormous development that parallels the growth in diversity of synthetic applications of such inorganic materials as catalysts, adsorbents, sensors, semiconductors, and antibacterial agents [77]. Micelles as nanoreactors have been applied for the fabrication of such inorganic particles are known to affect their morphology, particle size, and their final physicochemical properties considerably [78].

Hada et al. reported the preparation of nickel oxide nanoparticles (NPs) using a single microemulsion system by hydrolysis of nickel chloride hexahydrate [75]. The mixed reverse microemulsion was prepared by stirring cyclohexane, isopropanol, aqueous solution of nickel salt with a mixture of Tween 80 with sodium dioctyl sulfosuccinate (AOT), and an anionic surfactant at room temperature. Upon alkalinisation of the mixture, a nickel hydroxide precipitate was obtained. The favourable emulsification properties of Tween 80 results in stabilization of the surfactant film in the emulsion to a greater extent than previous reports with the use of conventional surfactants [79]. In turn, the use of such glycan-based amphiphiles serves to control the size and polydispersity of the prepared nanoparticles [80]. X-ray diffraction studies confirmed that a face-centred cubic pattern corresponding to a crystalline nickel hydroxide structure was produced. The overall process is shown in Figure 3. The spherical shape of the NPs was affirmed by SEM (scanning electron microscopy) results, where the particle size was estimated below 50 nm.

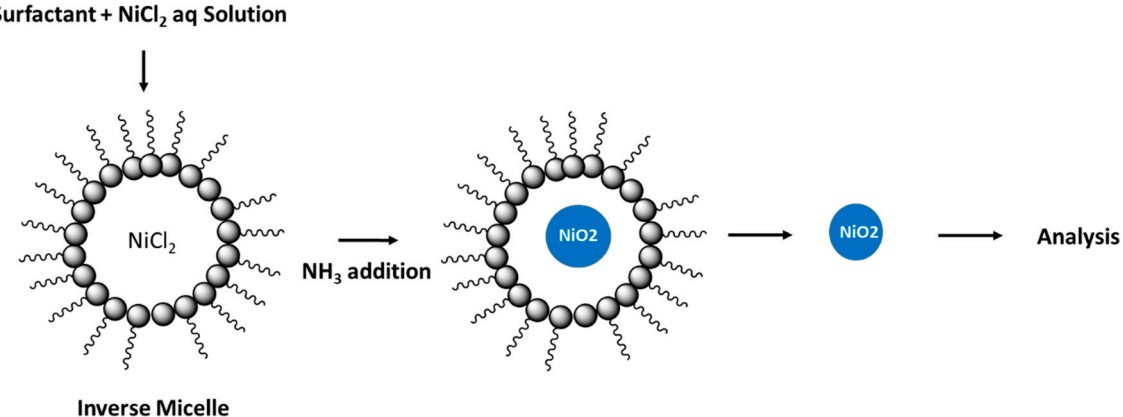

**Figure 3.** Overall process of nickel oxide formation in the micelle. Redrawn with modification from Ref. [75].

The nanoparticle size and other properties such as the morphology is affected by the type of surfactant used in the nanoreactor synthesis. The size of metallic modified $TiO_2$ nanoparticles prepared in the microemulsion aggregates was dependent on the type of surfactants used for preparation of the emulsion, along with other experimental parameters [81,82]. Doping of Au/Ag on titana enhanced its photocatalytic activity by 80% for photo degradation of phenolic compounds. The catalyst activity was strongly dependent on the size of the gold/silver particles that resulted upon the addition of the reducing agent (sodium borohydride) to a microemulsion system, which contained silver and gold precursors. Titanium oxide NPs were added to the system in the last stage. Three different types of surfactants were used for the microemulsion preparation, which included sodium dioctyl sulfosuccinate (AOT), Triton-X100, and Span-80 individually. A comparison of the results showed that size of the resulting NPs from the use of a glycan-based surfactant (Span-80) was ca. 20 nm, which are nearly one third in size for similar nanoparticles prepared in a microemulsion system using a conventional anionic surfactant system. X-ray diffraction (XRD) patterns of the samples provided evidence of gold and silver on the surface of the titana particles. Photodegradation of phenol under visible light in the presence of Au/Ag-$TiO_2$ NPs was enhanced considerably, in comparison to naked titana. The proposed mechanism of photocatalytic degradation is shown in Figure 4. It was posited that the presence of additional metal on the surface of titana particles facilitated the process of electron transfer, which resulted in greater photocatalytic activity.

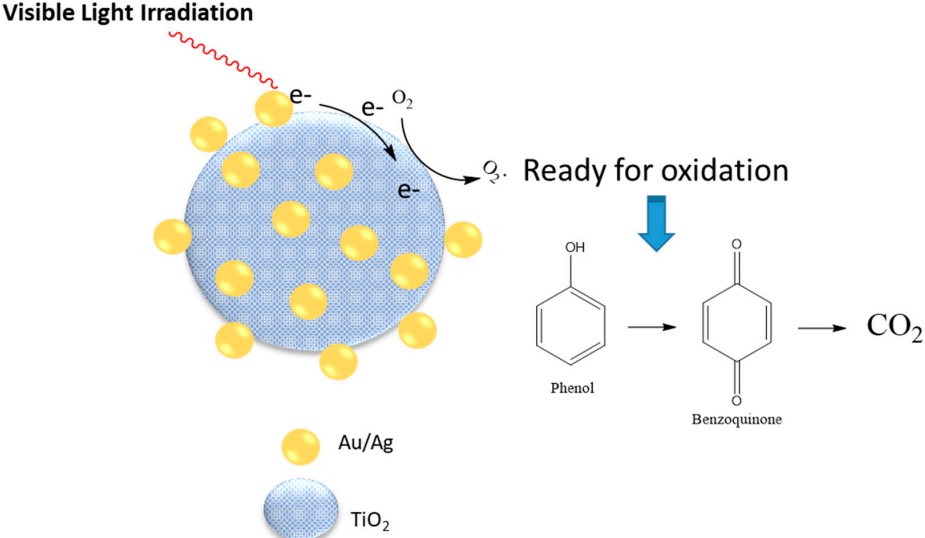

**Figure 4.** Proposed mechanism for photodegradation of phenol using Au/Ag $TiO_2$ as a catalyst. Redrawn with modification from Ref. [81].

In another study, anatase ($TiO_2$) was synthesized by micellar nanoreactors using Tween 80 as the surfactant. The size of the NPs was controlled by the size of the micelles. The same system was used to prepare hematite ($Fe_2O_3$) in the presence of carbon nanotubes as the inducer [83]. Additionally, the hydrolysis of titanium tetraisopropoxide in an emulsion system containing water, toluene, and a mixture of Span and Tween 80 was reported for the preparation of such nano-$TiO_2$ powders [84]. The ratio of water to surfactant was evaluated as the main factor that affected the size and physical properties of the NPs. A study of the transmission electron microscopy (TEM) images showed the formation of spherical particles with a narrow size distribution near 50 nm. The NPs were calcined and their pore sizes were estimated to be ca. 10 nm, according to a BET analysis of nitrogen adsorption isotherms.

Pineda-Reyes and coworkers illustrated the formation of zinc oxide NPs by a microemulsion system [85]. ZnO was prepared using zinc acetate as the precursor and sodium hydroxide as the precipitating agent in a microemulsion consisting of water, oil, and a mixture of Span and Tween 80 surfactants. The NPs were calcined to make a powder with a homogeneous size distribution with particle sizes of 14 nm, which was confirmed by TEM and characterized by complementary XRD measurements.

In addition to the above-mentioned examples, other studies have reported the synthesis of inorganic compounds in nanoreactors. Zhang et al. provided an account of the formation of CdS nano dots in a polymeric colloidal nanoreactor made using sorbitan oleate as the glycan-based surfactant. CdS nano dots were prepared and used for photocatalytic degradation of methylthionine chloride [86]. The synthesis of nickel sulphide NPs in the microemulsion, which contained tetradecane, water and sucrose ester as the surfactant was reported as a simple method to make monodisperse and regular-shaped particles [87]. Monodisperse $SiO_2$-$TiO_2$ ceramic NPs were prepared in a nano emulsion system that contained a mixture of water, decane, and Span-Tween 80 to obtain the required HLB of the system, where a ceramic alkoxide was used as the precursor [57]. The stability of the nano emulsions prevented Ostwald ripening and deleterious effects related changes in the droplet size.

### 2.2. Polymerization

Although the application of nanoreactors for free-radical polymerization [88] is well known, the utility of glycan-based surfactants as nanoreactors is sparsely reported in the field of polymer synthesis. Whereas the view that "the best solvent is no solvent" is widely appreciated, especially from a practical chemical synthesis perspective. The practical feasibility of running chemical reactions at a larger scale without solvent is a challenging task, mainly due to the key role of heat and mass transfer effects [89].

Fatty acid glucamides were investigated as emulsifiers for emulsion polymerization, where these materials can replace toxic surfactants such as alkyl benzene sulphate or nonylphenol ethoxylates [90]. It was reported that vinyl monomers were polymerized in a stable emulsion system using persulfate as an initiator to prepare the polymer in its dispersed form. The same method was used for the polymerization of butyl acrylate (BA) and methyl methacrylate (MMA) in the presence of a mixture of APG and an anionic surfactant in a semicontinuous seeded emulsion polymerization reaction (*cf.* Figure 5) [91,92]. The random copolymers prepared by this method were characterized by particle size analysis, where the results reveal a unimodal distribution of small particles. Reaction parameters like the mass-ratio of the surfactants to the total reaction mixture, such as the ratio of BA to MMA, and the amount of initiator were changed to ascertain the most influential parameters on the particle size distribution.

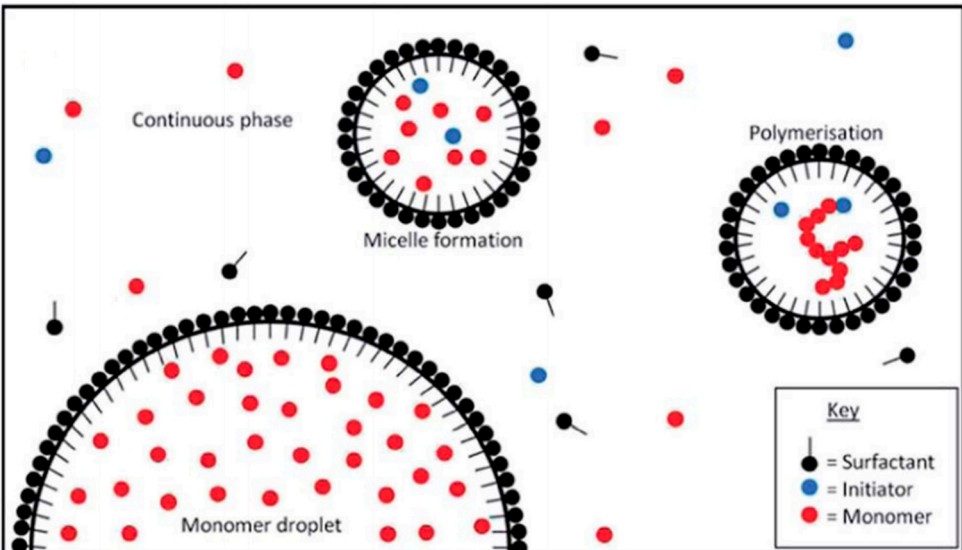

**Figure 5.** Emulsion polymerisation of acrylates. Adapted from Ref. [92], Published by the Royal Society of Chemistry (RSC).

Acrylamide-based polymers prepared by inverse nano emulsions were investigated using a mixture of surfactants that include a block copolymer and Tween-61. This combination of amphiphiles yield stable emulsions that prevent the emulsion separation and formation of oil spills during emulsion polymerization [93]. The emulsion polymerization process was started by activation of an initiator in the micelles, whereupon completion of the reaction, the resulting polymer system was cooled down to quench the reaction process. The polymers prepared by this approach have specific applications for industry. For example, high molecular weight polymers or copolymers of acrylamides of this category are widely used in wastewater treatment and for paper production [94].

*2.3. Synthesis of Organic Compounds*

The synthesis of small organic molecules was studied in nanoreactors and compared with reactions in bulk solution, where nanoreactor synthesis was achieved at shorter time with greater efficiency overall [95]. These reports were contrary to the previous precept that starting materials do not react with each other when dissolved in a solvent [96]. Nanoreactors can yield highly concentrated media with stereo-, regio-, and enantio-selective properties [97–99]. Hydrophobic effects play an essential role for the orientation of precursor materials in aqueous media [100]. In the micellar phase, the concentration of reactants is much higher when compared to results obtained for bulk solution [101]. As well, nanoreactor conditions contribute to restricted molecular movement since the presence of micelles may enforce regioselectivity effects [102].

Glucamide is a $C_{12}$ fatty acid that has been used to facilitate equimolar esterification reactions of benzoic acid and methanol in water under a range of conditions [103]. The authors also investigated their proposed method for a variety of carboxylic acids including aromatic and aliphatic moieties to confirm the efficiency of micellar system for such esterification reactions. The studies involved micellar systems even after recycling and reuse of the glycan-based surfactant over multiple reaction cycles.

Ge and coworkers [53] designed a bifunctional glycan-based surfactant to investigate its micellar structure and effects of chelation on a carbon-nitrogen/sulphur coupling reaction. The Cu-catalyzed reaction is a crucial way to make C-N/S bond in organic chemistry [104], where the main disadvantage of this process involves the use of toxic organic solvents like DMSO or dioxane. The replacement of these solvents with water makes it possible to move toward a greener reaction pathway. The authors reported that the micellar size of N-alkyl lactosamine as a glycan-based surfactant was changed after addition of halobenzene to the aqueous media. The effect was attributed to the

encapsulation of the reactant in the core of the micelles while the Schiff base functionality on the corona of the micelles' was chelated with a copper salt and its close proximity to the water-soluble nucleophile. The authors revealed that the reaction occurred in the palisade region of the micelle where all of the reactants are localized (*cf.* Figure 6). The coupling process was completed through a reductive elimination reaction.

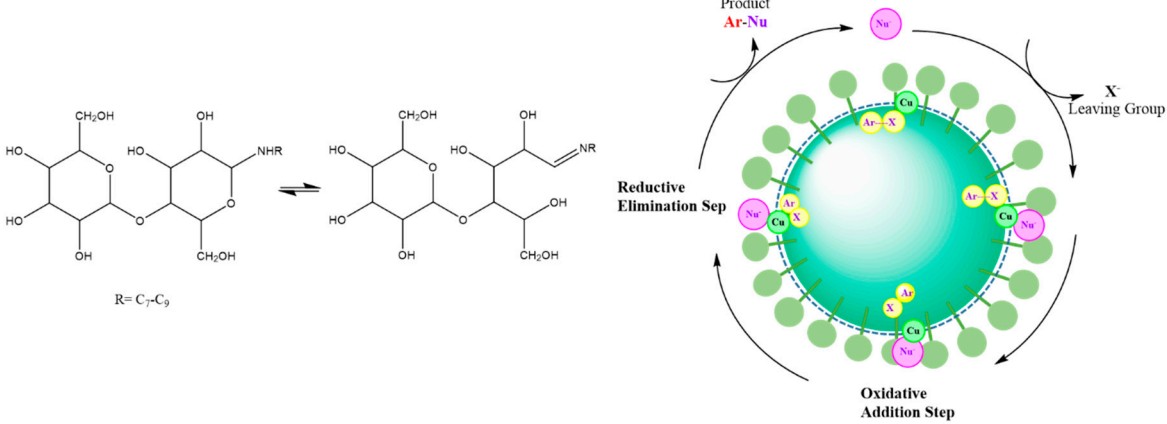

**Figure 6.** Left Side: chemical structure of used glycan-based surfactant; and Right Side: Schematic representation of the proposed mechanism of C-X coupling reaction in micelles of glycan-based surfactant. Redrawn from Ref. [53].

In a similar study, crocin, a natural glycan-based surfactant, was used to accelerate a redox reaction for a citric acid-KMnO$_4$ system. The authors reported a 5-fold increase in the rate of oxidation of citric acid in the presence of crocin. The effect was explained by incorporation of citric acid and permanganate (MnO$_4^-$) species in the Stern layer of the micelles that led to an enhanced reaction rate [105].

The importance of enantioselective reactions in micellar media relates to their application as models to mimic enzyme catalysis and asymmetric synthesis [99]. A unique study carried out by Kida demonstrated that the enantioselective hydrolysis of α-amino acid esters in micelles containing different types of fatty acid glucamides [106]. They described the role of structural variation of the surfactants on the hydrolysis rate and enantioselectivity of the hydrolysis reaction. The structure of the hydrophobic moiety of the surfactant, including alkyl chain length and number of alkyl chains, along with the stereochemistry of the sugar units' linkages was shown to affect the enantioselectivity of the reaction. The same reaction that used Triton X-100 as the surfactant did not show evidence of the role of enhanced enantioselectivity for the hydrolysis reaction.

In a study by Kumar et al., they revealed that the chiral interior of the micellar assembly comprised of a D-glucose based surfactant would provide a suitable environment for a cycloaddition reaction with high regioselectivity in water as illustrated in Figure 7 [107]. It was suggested that the intrinsic chirality of the D-glucose based surfactant was posited to play an important role in directing the stereochemistry of the reaction. They extended this methodology for the reaction between a series of aromatic aldehydes with phenyl hydroxylamine, where the reaction time, yield, regio-, and stereo-selectivity were affected by the presence of such glycan-based surfactants. The concentration of surfactants was above their CMC to ensure micelle formation. Upon completion of the reaction, the product was extracted from the organic phase while the aqueous phase containing the surfactant was reused for up to five consecutive cycles.

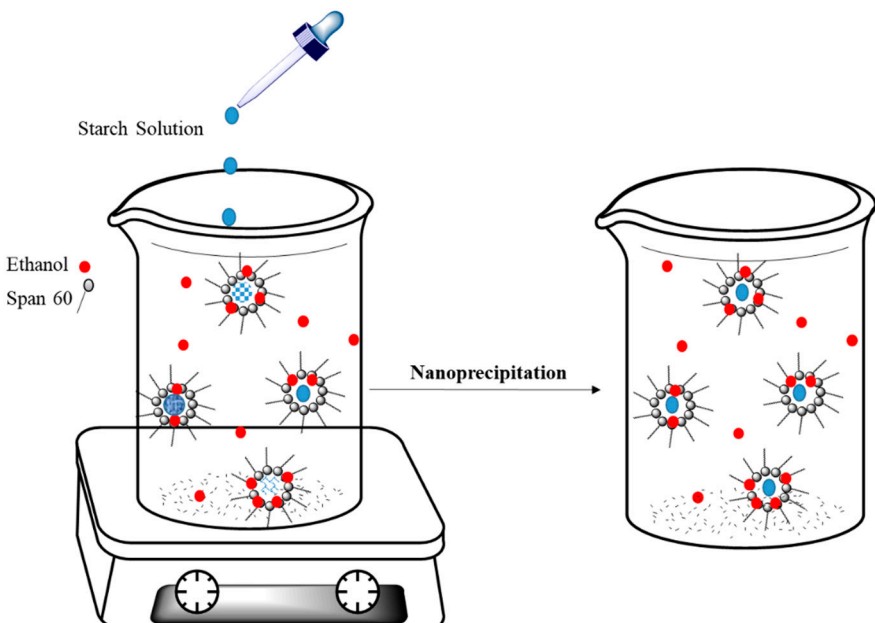

**Figure 7.** A cycloaddition reaction carried out in a D-glucose based surfactant system. Redrawn from Ref. [106].

### 2.4. Other Applications

There are various reports on the miscellaneous application of nanoreactors like crystallization of a pigment [108], which affect particle size, due to a confined space provided for crystallization. The pigment was prepared in the droplets containing Span-80 as the emulsifier of the emulsion system. The resulting pigment had a relatively narrow and uniform particle size distribution, as compared with crystallization products from the bulk solution. Furthermore, the emulsion-based pigment synthesis showed higher UV–Vis absorbance and better water dispersion, as compared with pigments prepared in bulk solution.

The synthesis of biopolymer NPs using nanoreactors for use as the main components in colloidal delivery was previously reported in the literature [109]. Biopolymers can undergo precipitation within micelles when they encounter a precipitating agent. In the case of starch nanoparticles, the biopolymer will form a precipitate in reverse micelles of Span-60 upon interaction with ethanol, as shown in Figure 8 [110]. This strategy was used for encapsulation of curcumin in starch nanoparticles. Chin and coworkers reported that encapsulation took place if curcumin was added to the oil-phase prior to induction of precipitation of starch nanoparticles [111]. Kobiasi et al. [112] also reported the preparation of cross-linked chitosan nanoparticles in micelles prepared by the use of different glycan-based surfactants that include Tween 80, Span 80, and Span 85. The aim of the study was to elucidate the effects of micelle structure on the properties of the nanoparticle products. In turn, the nanoparticles were used for in vivo biomedical studies directed toward cellular uptake and trafficking to lymph node sites [112].

**Figure 8.** Preparation of starch-based nanoparticles in Span-60 micelles. Redrawn with modification from Ref. [110].

In a recent study, a unique application of nanoreactor systems was developed by Su and coworkers [113], where they synthesized magnetic hybrid nanogels with unique superparamagnetic properties. The use of dextran imparts biocompatibility with other

biomaterials, favouring its utility as a suitable biomedical probe of materials for magnetic resonance imaging (MRI) studies [114]. Nanogels that contain magnetic iron oxide NPs encapsulated in a dextran polymer matrix were prepared from physical doping of water-soluble iron oxide ($Fe_3O_4$) NPs in droplets of Schiff base mediated dextran that employed the nanoreactor technique. In the first step of the process, the NPs were prepared and mixed with an oxidized dextran solution. Then, the dextran-based nanogels were chemically cross-linked with ethylene diamine in the water droplets of a water-in-oil emulsion that consisted of cyclohexane and water with Span 80 and Tween 80 as the emulsifiers. The schematic representation of the synthetic method is illustrated in Figure 9. Magnetization studies showed that the prepared magnetic nanoparticles behaved like iron oxide nanocrystals while their spin–spin relaxation time ($T_2$) was longer than conventional magnetic particles, a property that offers promising potential for application as MRI probe systems.

**Figure 9.** Preparation of a dextran-based magnetic resonance imaging (MRI) probe. (**a**) Preparation of iron oxide, (**b**) aldehyde dextran fabrication, (**c**) blending of an aldehyde form of dextran with magnetic nanoparticles in a nanoreactor assembly, and (**d**) cross-linking by ethylene diamine and washing with ethanol to remove surfactants. Redrawn with modification from Ref. [113].

## 3. Conclusions

This review focused on a coverage of recent advances in applications of glycan-based surfactant emulsions as nanoreactor constructs. The glycan-based micelles are considered as a unique reactor systems for various chemical reactions that involve encapsulation of the starting materials and products due to their amphiphilic and colloidal properties. Nanoreactor constructs have emerged as a versatile reaction media for the synthesis of diverse chemical species such as inorganic nanoparticles, organic compounds, and advanced polymer materials. In this mini-review, a brief overview of oil-in-water or water-in-oil emulsions as nanoreactors comprised of low to medium molecular weight glycan-based amphiphiles were described and their synthetic applications were highlighted. The synthesis of the glycan-based surfactants is generally facile overall, as revealed by reactions with relatively few steps and high yield. The product properties such as the size and morphology in the case of inorganic nanoparticles, along with enantiomeric excess or regioselectivity in case of organic compounds, and particle size for biopolymer nanoparticles are shown to depend directly on the micellar (nanoreactor) structure [33]. The synthetic advantages of nanoreactors are combined with their outstanding emulsification properties, biodegradability, eco friendliness, safety, and competitive material cost [115]. From a technological perspective, the selection of amphiphiles with variable glycan head groups and lipophilic fragments can yield a wide array of aggregate morphology (i.e., spherical micelles, bilayers, worm-like aggregates, etc.), which can ultimately control the thermodynamics and kinetics of reactions, along with nanoreactor physicochemical properties of the reaction products prepared by this approach [116]. Moreover, the constitutional inner cavity chirality of nanoreactors is a unique and impressive feature, which is mainly considered for the synthesis of organic compounds or preparation of chromatographic systems [117].

The further development of this field of low to medium molecular weight glycan-based amphiphiles will contribute to the greater utilization of polymer-based glycan amphiphiles, especially as researchers develop an improved understanding of structure–function relationships for such colloidal materials in various media [118]. Another interesting prospect for these green and biocompatible surfactants is the role of surfactant–polymer interactions. Recently these mixtures have gained attention due to their potential utility in drug delivery, catalysis, etc., [119,120] and the utility of such glycan-based systems as alternatives to conventional petrochemical-based surfactants.

**Author Contributions:** Conceptualization, B.V.; Investigation, B.V.; Resources, L.D.W.; Writing—Original Draft Preparation, B.V.; Writing—Review and Editing, B.V. and L.D.W.; Visualization, B.V.; Supervision, L.D.W.; Funding Acquisition, L.D.W. All authors have read and agreed to the published version of the manuscript.

**Funding:** This research was funded by the Government of Canada through the Natural Sciences and Engineering Research Council (NSERC), Discovery Grant Number: RGPIN 2016-06197. The financial support provided by the University of Saskatchewan in the form of a Dean's Scholarship for B.V. is gratefully acknowledged. The APC was funded by MDPI as part of a special invitation to the corresponding author (L.D.W.).

**Institutional Review Board Statement:** Not applicable.

**Informed Consent Statement:** Not applicable.

**Data Availability Statement:** The data presented in this study are available within the article.

**Acknowledgments:** B.V. acknowledges Leila Dehabadi, Henry Agbovi and Chen Xue for helpful discussions.

**Conflicts of Interest:** The authors declare no conflict of interest.

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
