# Peer review of "A Review on Recent Progress of Glycan-Based Surfactant Micelles as Nanoreactor Systems for Chemical Synthesis Applications"

_2673-4176, doi:10.3390/polysaccharides2010012_

Round 1
Reviewer 1 Report
The article studies the nanoreactor concept and its application as a modality to carry out chemical reactions in confined and compartmentalized structures. The aggregation of glycan-based surfactants to afford micelles and the synthesis of selected classes of reactions by the nanoreactor technique are discussed.
The article has publishable material and my comments are as follows:
- More physics of results is required.
- Reference list is not uniform. Author should use ISO abbreviation for Journals names. It should be as per required style of journal.
- Improve the legend and quality of figures.
- All equations should be labeled in sequential order and their numbers should be aligned.
- There are few language and typographic errors and some "glitches" of editing must be fixed in revised paper.
- Compare the results with previous work for validation.
- For fortifying the introduction section with the new publications, old references should be replaced with new ones.
Accordingly, I advise that the manuscript is put under minor revision until the inquires addressed are responded to.
Author Response
Reviewer Comments on MS ID: polysaccharides-1091730
Reviewer #1
- More physics of results is required.
Response: Further edits were added on pages 6-8 in the revised manuscript.
- Reference list is not uniform. Author should use ISO abbreviation for Journals names. It should be as per required style of journal.
Response: The references were edited to address the reviewer comments.
Improve the legend and quality of figures.
Response: The corresponding edits were made to address the reviewer concerns.
- All equations should be labeled in sequential order and their numbers should be aligned.
Response: The corresponding edits were made to address the reviewer concerns.
- There are few language and typographic errors and some "glitches" of editing must be fixed in revised paper.
Response: The manuscript has been comprehensively edited to address the reviewer concerns.
- Compare the results with previous work for validation.
Response: Comparison of the results was carried out to address the reviewer concerns
- For fortifying the introduction section with the new publications, old references should be replaced with new ones.
Response: The introduction was edited to address the reviewer concern.
In summary, the authors wish to acknowledge Reviewer #1 for the insightful comments and constructive criticism along with the opportunity to improve the overall quality of this manuscript submission.

Reviewer 2 Report
The current manuscript compiles the most recent applications of micellar and emulsion systems formed by glycan-based surfactant, a type of "green" class of surfactants that have attracted great attention in the last years. Although several manuscripts describe the preparation and fundamental characterization of a variety of these amphiphiles, a striking absence of data on potential applications is seen in the literature and this review will partially fill this gap. However, several issues have to be addressed in the current version of the manuscript, which, once solved, will increase its suitability for publication in the journal. Hence, I recommend its acceptance after major revisions, as stated below:
Minor english editing is needed in some parts of the text, thus making necessary a review in the whole manuscript
In topic 1.1, information on the different biodegradability, or even degradability upon chemical conditions, of the exemplified classes of glycan-based surfactants would be interesting. It is quickly mentioned that glucamides are less prone to be hydrolyzed than sorbitan fatty acids, but this could be expanded on.
In topic 1.2.1, the statement that at around the cmc the water-air interface is saturated by surfactant molecules, although highly reproduced in the literature, is wrong. Surfactants stop going to the interface because at this particular concentration a new energetically favourable path is available: aggregate into micelles. Less work is done by moving the surfactant molecules from the bulk solution to the interior of the micellar aggregate than moving them from the solution to the interface. This information can be found in colloid text books. Please, correct that (and also Figure 1).
Also in topic 1.2.1, data on the equilibrium surface tension of some representative glycan-based surfactants could be added in Table 1. It is hard to find such a compilation for this type of surfactants. Thus, this information would increase the attention of potential readers of the manuscript.
In topic 1.2.2, especially in lines 160-162, DLS can give the hydrodynamic diameter or hydrodynamic radius (which is 1/2 of the diameter), depending on what the researcher wishes. So, saying that "The diameter, which is determined by DLS, is the hydrodynamic diameter (Rh)" is really confusing. Please, rephrase this. Also, from my own experience, most surfactant solutions give rise to very poor scattering intensities (due to their extremely low cmc values and thus micelle concentration), thus making DLS a technique rarely used to estimate micelle size. This should be considered in the new version of the manuscript.
Data for sucrose decanoate displayed in Table 1 contradicts information previously provided in topic 1.1.4 which says that sucrose esters with alkyl chains below 12 carbon atoms do not present surface activity (line 132). This statement has to be revised and rewritten, considering the available literature on this class of surfactants.
Text in topic 2.1, line 221, states that 1982 was less than 20 years ago (!).
In the same topic, text in line 234 states that Tween 80 stabilizes micelles, but I think in this case, it stabilizes the surfactant film in the emulsion (no micelle is supposed to exist but a reverse emulsion) which brings me the concern that authors are confusing micelles (equilibrium, that is thermodynamically stable, solutions) with emulsion (a kinetic stabilized dispersed system). Following the text, in the next paragraph (line 253), "micelles of a microemulsion" are refereed again. Same thing in line 309, 342, 421, and Figure 5.
It would be important to emphasize that the work of Zhang et al., described in line 301 and so, uses a glycan-based surfactant as emulsifier.
In conclusions, line 489, not only glycan-based surfactant micelles were reported in the review, but also glycan-based surfactant emulsions. Please, rephrase that.
References 35 and 41 are the same paper. Please, correct this. There is also a problem with refs. 49 and 50 (same title, different authors).
As a final comment, the interaction of surfactants with polymers is of great importance, since both components appear in almost every formulation (including food, cleaning, pharmaceutics and cosmetic products). More recently, the formation of liquid crystalline nanoparticles by surfactant-polymer interactions gained much attention due to their potential use in drug delivery, catalysis etc. (see the works of Watson Loh - a review in Current Opin Colloid Interf Sci 2017 and a paper in Adv Mat Interf 2019, for example). However, all studies comprise petroleum-based surfactants. Hence, the use of "green" surfactants for this kind of study is nule in the literature and I see a great opportunity to use glycan-based surfactants in this field, due to their eco-friendless and biocompatibility. A comment in this sense could strengthen the concluding remarks.
Author Response
Reviewer Comments on MS ID: polysaccharides-1091730
Reviewer #2
Minor english editing is needed in some parts of the text, thus making necessary a review in the whole manuscript
Response: The manuscript has been comprehensively edited for language, syntax, and clarity to address the reviewer concerns.
In topic 1.1, information on the different biodegradability, or even degradability upon chemical conditions, of the exemplified classes of glycan-based surfactants would be interesting. It is quickly mentioned that glucamides are less prone to be hydrolyzed than sorbitan fatty acids, but this could be expanded on.
Response: Edits to address the reviewer comments was carried out as outlined on page 3 of the revised manuscript.
In topic 1.2.1, the statement that at around the cmc the water-air interface is saturated by surfactant molecules, although highly reproduced in the literature, is wrong. Surfactants stop going to the interface because at this particular concentration a new energetically favourable path is available: aggregate into micelles. Less work is done by moving the surfactant molecules from the bulk solution to the interior of the micellar aggregate than moving them from the solution to the interface. This information can be found in colloid text books. Please, correct that (and also Figure 1).
Response: The authors have addressed the reviewer concern in the revised version of the manuscript, as outlined on pages 4-5.
Also in topic 1.2.1, data on the equilibrium surface tension of some representative glycan-based surfactants could be added in Table 1. It is hard to find such a compilation for this type of surfactants. Thus, this information would increase the attention of potential readers of the manuscript.
Response: To address the reviewer concern, the authors have made edits on page 8 of the revised manuscript.
In topic 1.2.2, especially in lines 160-162, DLS can give the hydrodynamic diameter or hydrodynamic radius (which is 1/2 of the diameter), depending on what the researcher wishes. So, saying that "The diameter, which is determined by DLS, is the hydrodynamic diameter (Rh)" is really confusing. Please, rephrase this. Also, from my own experience, most surfactant solutions give rise to very poor scattering intensities (due to their extremely low cmc values and thus micelle concentration), thus making DLS a technique rarely used to estimate micelle size. This should be considered in the new version of the manuscript.
Response: To address the reviewer concerns, corresponding edits were applied as outlined on pages 5-7 of the revised manuscript.
Data for sucrose decanoate displayed in Table 1 contradicts information previously provided in topic 1.1.4 which says that sucrose esters with alkyl chains below 12 carbon atoms do not present surface activity (line 132). This statement has to be revised and rewritten, considering the available literature on this class of surfactants.
Response: To address this point, the reviewer concern was address in the revised version of the manuscript.
Text in topic 2.1, line 221, states that 1982 was less than 20 years ago (!).
Response: We have addressed the concern raised by the reviewer.
In the same topic, text in line 234 states that Tween 80 stabilizes micelles, but I think in this case, it stabilizes the surfactant film in the emulsion (no micelle is supposed to exist but a reverse emulsion) which brings me the concern that authors are confusing micelles (equilibrium, that is thermodynamically stable, solutions) with emulsion (a kinetic stabilized dispersed system). Following the text, in the next paragraph (line 253), "micelles of a microemulsion" are refereed again. Same thing in line 309, 342, 421, and Figure 5.
Response: The reviewer comments were addressed in the revised manuscript.
It would be important to emphasize that the work of Zhang et al., described in line 301 and so, uses a glycan-based surfactant as emulsifier.
Response: We have made corresponding edits on page 10 in the revised manuscript.
In conclusions, line 489, not only glycan-based surfactant micelles were reported in the review, but also glycan-based surfactant emulsions. Please, rephrase that.
Response: The corresponding correction was made to address the reviewer comment.
References 35 and 41 are the same paper. Please, correct this. There is also a problem with refs. 49 and 50 (same title, different authors).
Response: The references were corrected accordingly.
As a final comment, the interaction of surfactants with polymers is of great importance, since both components appear in almost every formulation (including food, cleaning, pharmaceutics and cosmetic products). More recently, the formation of liquid crystalline nanoparticles by surfactant-polymer interactions gained much attention due to their potential use in drug delivery, catalysis etc. (see the works of Watson Loh - a review in Current Opin Colloid Interf Sci 2017 and a paper in Adv Mat Interf 2019, for example). However, all studies comprise petroleum-based surfactants. Hence, the use of "green" surfactants for this kind of study is nule in the literature and I see a great opportunity to use glycan-based surfactants in this field, due to their eco-friendless and biocompatibility. A comment in this sense could strengthen the concluding remarks.
Response: The corresponding updates to the revised manuscript were carried out to address the reviewer comments.
In summary, the authors wish to acknowledge Reviewer #2 for the insightful comments and constructive criticism along with the opportunity to improve the overall quality of this manuscript submission.

Round 2
Reviewer 2 Report
The authors have addressed all the issues raised in the first round of review. The manuscript will greatly contribute to the field and be of interest to the readers. The text, at the current stage, can be published with no need of extra revisions.